# Enhanced Antioxidant and Anticancer Potential of *Artemisia carvifolia* Buch Transformed with *rol A* Gene

**DOI:** 10.3390/metabo13030351

**Published:** 2023-02-27

**Authors:** Amna Naheed Khan, Erum Dilshad

**Affiliations:** Department of Bioinformatics and Biosciences, Faculty of Health and Life Sciences, Capital University of Science & Technology, Islamabad 44000, Pakistan

**Keywords:** *Artemisia carvifolia* Buch, *Agrobacterium tumefaciens*, *rol* gene, flavonoids, phenylalanine ammonia-lyase, chalcone synthase, antioxidant assays, antiproliferative activity

## Abstract

Secondary metabolites have been shown to possess a range of biological functions. Flavonoids, due to their ability to scavenge ROS, are famous antioxidants. The plants of *Artemisia* species are rich sources of flavonoids; however, the amount of these metabolites is less. In the current study, the flavonoid content was detected and then enhanced by genetically modifying the *Artemisia carvifolia* Buch with *Agrobacterium tumefaciens* strain GV3101 carrying *rol A* gene. The transformation of *rol A* gene was confirmed with PCR and the gene copy number was confirmed by Southern blot analysis. The HPLC analysis revealed the presence of catechin (3.19 ug/mg DW) and geutisic acid (2.22 ug/mg DW) in transformed plants, unlike wild-type plants. In transformed plants, all detected flavonoids (vanillic acid, rutin, catechine, gallic acid, syringic acid, caffeic acid, coumaric acid, geutisic acid, ferulic acid, and cinnamic acid) were increased up to several folds. Real-time qPCR revealed the higher expression levels of the genes for flavonoid biosynthesis enzymes phenylalanine ammonia-lyase *(PAL)* and chalcone synthase *(CHS)* in plants transformed with *rol A* genes, as the expression levels were increased up to 9–20-fold and 2–6-fold, respectively. The *rol A* transgenic lines T3 and T5 carrying two copies of *rol A* gene, particularly showed higher expression of both *PAL* and *CHS* gene, with the highest expression in T3 line. The transgenic lines demonstrated an average increase of 1.4-fold in the total phenolic content and 1–2-fold in the total flavonoid content as compared to wild-type plants. Total antioxidant capacity and total reducing power were increased up to an average of 1–2-fold and 1.5–2-fold respectively, along with increased free radical scavenging ability. Furthermore, the *rol A* gene transgenics were found to have much greater cytotoxic capacity than the *A. carvifolia* wild-type plant against the MCF7, HeLA, and HePG2 cancer cell lines. Current findings show that the *rol A* gene effectively increases the flavonoid content of *A. carvifolia* Buch, boosting the plant’s capacity as an antioxidant and an anticancer. This is the first-ever report, demonstrating the genetic transformation of *Artemisia carvifolia* Buch with *rol A* gene.

## 1. Introduction

Now a days naturally occurring compounds and their associated medicines are used to treat nearly 87% of human diseases, including different bacterial infections, cancer, and immunological disorders. It has been revealed that almost 25% of approved drugs around the world are obtained from plant origin. Furthermore, about 80% people in many developing countries use medicines obtained from plants for their health-related issues [1]. *Artemisia* L. is a diverse genus of fragrant small herbs and shrubs widely spread in northern temperate regions. The *Artemisia* L. genus comprises over 500 species, found in the northern hemisphere, in Asia, Europe, and North America. It belongs to the important family Compositae (Asteraceae) [2].

Plant extracts attained from *Artemisia* species have pharmacological properties and are used for the treatment of illnesses such as depression, anxiety, insomnia, epilepsy, stress, irritability and psychoneurosis. Additionally, plants from *Artemisia* species hold a wide range of biological functions such as antibacterial, antitumor, hepatoprotective, antimalarial, antiseptic, antispasmodic, and antirheumatic activities [1,3]. The primary cause of the intense and aromatic scent of some *Artemisia* species is high quantities of volatile terpenes, which are components of their essential oils and are found in particular in the leaves and flowers of these plants. Numerous species of the *Artemisia* genus from around the world have undergone extensive chemical analysis to determine the chemical makeup of essential oils. Numerous investigations have revealed that the terpene components of the essential oils produced by *Artemisia* species exhibit notable intraspecific differences [2].

A class of phenylpropanoids with low molecular weight compounds known as flavonoids are found in plant cells vacuoles where they act as water soluble pigments [4]. Due to their capacity to scavenge reactive oxygen species (ROS), flavonoids are regarded as antioxidants or phytoalexins [5]. Thus, they shield plants against both biotic and abiotic factors—including UV exposure, extreme cold temperatures, microbial contamination, and insect feeding—to prevent serious harm [6,7,8]. Additionally, serving as chemical messengers in plants, flavonoids draw pollinating insects and play a role in the metabolism of auxin [9]. The genus *Artemisia* is a plentiful supplier of flavonoids [10]. Although *A. annua*—from which over 50 flavonoids have been found—has received the most attention among the *Artemisia* species, other species, such as *A. absinthium* L. [11], *A. asiatica* [12], and *A. herba-alba* [13] have also been found to contain a number of flavonoids.

However, the amount of these flavonoids in plants, besides this, at different growth stages of a plant fluctuations or variability in flavonoids concentrations has been observed. For example, the highest concentrations of chrysoplenetin and casticin were discovered in the leaves and flowers of one cultivar of *A. annua* during blossom [13]. Similar to this, a recent study found that the antioxidant capacity varied between cultivars, indicating that the antioxidant components, particularly flavonoids, can vary in content [10].

In order to improve the synthesis of secondary metabolites in plants, genetic engineering has proven to be the most effective method among various approaches by altering the biochemical pathways that are involved in the synthesis of secondary plant chemicals [14]. It has been revealed in different studies that *rol* genes act as a powerful inducers or activators of secondary metabolite production in various plant families [15]. The *rol A* gene encodes a growth-stimulating protein that binds to DNA, while the *rol B* gene regulates the signaling pathway of auxin by acting as a tyrosine phosphatase [16,17]. Besides this, the transformation of *Vitis amurensis* with *rol B* gene has increased the resveratrol synthesis [18]. It was further found that the anthraquinones production was increased with *rol B* in *Rubia cardifolia* transformed calli [19]. In plants as well as cell cultures that have undergone transformation, the *rol C* genes cytokinin glucosidase activity can activate the creation of a wide collection of secondary compounds, including ginsenosides and various alkaloids [20,21,22,23,24].

In the present study, flavonoids were detected in the *Artemisia carvifolia* Buch wild type and then flavonoid content was enhanced by genetically modifying *A. carvifolia* with *rol A* gene. Aimed to correlate the levels of gene expression responsible for the production of flavonoids with metabolite concentrations, real-time qPCR was used to analyze their response. For this purpose, two genes that encode *PAL* and *CHS* of the phenylpropanoid route of flavonoid biosynthesis were investigated. Furthermore, HPLC analysis was used to quantify the flavonoids in transformed and untransformed plants. Moreover, antioxidant potential and cytotoxicity were also tested. This is the first-ever report, demonstrating the genetic transformation of *Artemisia carvifolia* Buch with *rol A* gene.

## 2. Materials and Methods

### 2.1. Seed Germination and DNA Barcoding

For the present research, seeds of *Artemisia carvifolia* were utilized. These seeds were procured from Astore located at 74.8500° E, 35.3667° N and an elevation of 8500 feet in Northern Pakistan. Using 70% ethanol, the gathered seed surfaces were sterilized. After sterilization, a half-strength MS medium was used to germinate seeds. Extraction of plant genomic DNA from germinating plantlets was carried out according to a predetermined protocol [25]. DNA barcoding was used to identify *Artemisia carvifolia* buch. For this purpose, a non-coding region (spacer) between the *trnH* and *psbA* genes of chloroplast. Using primers of *psbA*: 5′-GTTATGCATGAACGTAATGCTC-3′; and *trnH*: 5′-CGCGCATGGTGGATTCACAATC-3′, DNA was amplified. The PCR reaction was conducted using the previously stated reaction parameters [26]. For PCR product purification, the Rapid PCR Purification System 9700 (Marligen Biosciences, Ijamsville, MD, USA) was employed. The PCR product was sequenced by utilizing the dideoxy-chain termination method. The BioEdit sequence alignment program was then used to identify and align the sequences.

### 2.2. Bacterial Strains and Plasmids

The plasmid pPCV002-A was carried by the *Agrobacterium tumefaciens* strain GV3101 that was graciously provided by Dr. A. Spena [27]. Its T-DNA region—which also includes the *NPTII* gene (*neomycin phosphotransferase* gene), the NOS (nopaline synthase) promoter, and terminator sequences—enables the CaM35S promoter to regulate the *rol A* gene’s expression. *Agrobacterium tumefaciens* (strain GV3101 carrying pPCV002-A) were introduced into a flask with LB and bacterial cultures were then left in the shaker incubator for the night at 120 rpm and 28 °C. The bacterial culture was utilized to transform the plants after 24 h of growth had been attained and OD had been evaluated using a spectrophotometer when it was between 0.2 and 0.8.

### 2.3. Transformation and Regeneration

The genetic transformation was achieved according to previously described method [22]. One-month-old nodal explants initially precultured on shooting media with 200 µM acetosyringone and hormone supplements comprising 0.5 mg/L BAP and 0.1 mg/L NAA were used for the experiment. The explants were infected with an *Agrobacterium tumefaciens* culture bearing the required constructs after three days. The explants were then placed on filter paper that was already autoclaved, for 1–2 min to eliminate excess bacterial culture and after that, they were put on MS shooting media enriched with 200 µM acetosyringone. The explants were kept in darkness at 28 °C. After two days, the explants were given three antibiotic-based washes before being placed on a media known as a selection medium with the same hormones and antibiotics concentrations as those previously reported [22]. After 3–4 weeks, regeneration of explants occurred so they were transferred to fresh selection medium, after that subculturing was carried out after every two weeks. The explants were then shifted to rooting media [22] for root development. The entire plants were then regenerated on selection media after four selection cycles.

### 2.4. Molecular Analysis

To confirm the transformation of *rol A* gene, molecular analysis was performed. For this purpose, the CTAB technique was used to isolate genomic DNA from 7 to 8 weeks old transformed and untransformed *Artemisia carvifolia* plants [25], and the alkaline lysis method was used to isolate plasmid DNA from *Agrobacterium* strain GV3101. A programmed DNA thermal cycler was used to perform the PCR analysis. The *rol A* gene forward primer 5′-AGAATGGAATTAGCCGGACTA-3′ and reverse primer 5′-GTATTAATCCCGTAGG TTTGTT3′ were utilized. The PCR conditions have been previously described [28]. Agarose gel of 1.5% w/v was prepared for agarose gel electrophoresis to confirm and analyze the PCR product.

### 2.5. Southern Blotting

For Southern blot testing of *rol A* transformed plants, 3–5 μg of plant genomic DNA was digested with EcoRI and then electrophoresed on 0.7% agarose gel. The processed DNA fragments are transferred to a nylon membrane which is positively charged, in accordance with the standard procedure [29]. As directed by the manufacturer, the DIG High Prime DNA Labeling and Detection Starter Kit II (Roche cat. no. 11585614910) was utilized to conduct the Southern blot analysis. The probe was created using *rol A* gene PCR products from plasmids that were labeled with digoxigenin (DIG)-11-dUTP and DIG High Prime DNA Labeling reagents. The probe was then hybridized with a membrane bearing genomic DNA at a temperature of 42–44 °C. The membrane was then placed on X-ray film to conduct the immunological detection process.

### 2.6. Evaluation of Flavonoids through an HPLC-DAD System

Polyphenols in extracts from both *A. carvifolia* plants of the transformed and wild types were measured using HPLC-DAD analysis. The protocol followed was reported earlier [30] with slight changes. For this experiment, dry sample extracts were diluted in methanol to provide a solution with 10,000 μg/mL concentration. Additionally, the stock solutions of 10 standards—including vanillic acid, rutin, catechine, gallic acid, syringic acid, caffeic acid, coumaric acid, geutisic acid, ferulic acid, and cinnamic acid—were prepared in methanol at 1000 μg/mL concentration. For the standard calibration curve, 10, 20, 50, 100, 150, and 200 μg/mL dilutions were made in series using these freshly prepared stock solutions. The amount injected was 20 μL, and constant flow rate of 1 mL/min was maintained. In mobile phase A, chemicals were combined with water in a ratio of 10:5:1:85 (methanol-acetonitrile-acetic acid-water), while in mobile phase B, only chemicals were used in a ratio of 60:40:1 (methanol-acetonitrile-acetic acid). In the gradient approach, the distribution of volume of B was as follows: 0–50% in 0–25 min, 50–100% in 25–30 min, 100% in 30–35 min. It was a room temperature HPLC analysis. Furthermore, UV retention periods, and absorption spectra of plant extracts were checked in comparison with standards in order to identify compounds. The column was always reconditioned for 10 minutes before the subsequent analysis began. By comparing retention rates of reference standards, peaks in the extracts were discovered. The analytes were identified using wavelengths and retention times that were specific for each metabolite as shown in Table 1.

### 2.7. Analysis of Genes Involved in the Flavonoid Biosynthetic Pathway Using Real-Time qPCR

In order to check the expression level of *rol A* gene, a semi-quantitative reverse transcriptase-polymerase chain reaction was performed with *rol A* gene primers as reported previously [22], using 1 μL of cDNA reaction mixture as a template. Following a previously described procedure, real-time qPCR was used to assess the expression of genes of the flavonoid biosynthetic pathway [22]. For amplification reaction, primers that target a certain gene were utilized. For this, the *PAL* forward and reverse primers were 5′-ACACTCGGTTAGCTATTGCTGCAA-3′ and 5′-CCATGGCGATTTCTGCACCT-3′, respectively. The *CHS* forward and reverse primers were 5′-AGGCTAACAGAGGAGGGTA-3′ and 5′-CCAATTTACCGGCTTTCT-3′, respectively. The actin primers used were 5′-ATCAGCAATACCAGGGAACATAGT-3′ (forward) and 5′-AGGTGCCCTGAGGTCTTGTTCC-3′ (reverse).

### 2.8. Antioxidant Potential Measurement

The antioxidant ability of both untransformed and transformed plants was assessed using in vitro antioxidant assays. The methodology used was same as previously reported in which one-gram air-dried powder of plant shoots of each type were used to prepare methanolic extract by soaking in 3 mL of methanol [31].

#### 2.8.1. Total Phenolic Content Measurement

Following a previously established procedure, plant extracts were measured for their total phenolic content (TPC) [32,33]. Gallic acid and DMSO were employed as positive and negative controls, respectively. TPC was represented as gallic acid equivalent.

#### 2.8.2. Total Flavonoid Content Measurement

The colorimetric method using aluminum chloride was used to assess the TFC of the under-researched plant extracts by using the previously described method [33]. Quercetin served as the positive control, and the negative control was DMSO. TFC was represented as quercetin equivalent.

#### 2.8.3. Total Antioxidant Capacity Measurement

Total antioxidant capacity (TAC) of the under-study plant extracts was calculated by using the reported methodology [34]. Ascorbic acid and DMSO were used as the positive and negative controls, respectively. TAC was represented as an ascorbic acid equivalent and determined by using the formula below:Ascorbic Acid Equivalence = 100/2.651 × Absorbance of sample μg/mL

#### 2.8.4. Total Reducing Power Measurement

The total reducing power of plant extracts from wild type and transgenic *A. carvifolia* was evaluated by following the reported methodology [35]. The appositive and negative controls were ascorbic acid and DMSO, respectively. The evaluation was carried out in triplicate. TRP was expressed as an ascorbic acid equivalent and quantified using the formula below:Ascorbic Acid Equivalence = 100/2.7025 × Absorbance of sample μg/mL

### 2.9. DPPH Free Radical Scavenging Assay

All plant extracts DPPH levels were calculated based on the reported protocol [31]. The experiments were performed in triplicate and the absorbance was measured at 517 nm. The percentage of DPPH free radical scavenging for each concentration of plant extract was determined by the following formula.
Percentage scavenging (%) = [1 − absorbance of extract/absorbance of control] × 100

### 2.10. Measurement of Anticancerous Activity through MTT Assay

The MTT test was applied to evaluate the antiproliferative activity of the plant extracts under investigation. This colorimetric method relies on the fact that living cells have the capacity to convert the tetrazolium dye MTT via mitochondria into its insoluble formazan, a distinctive purple precipitate [36]. Three cancer cell lines—HePG2 (derived from hepatic carcinoma), HeLA (derived from cervical cancer cells), and MCF7 (derived from breast carcinoma)—were employed for that aim. The samples were prepared and assay was performed by following the reported protocol [34]. Briefly, 1 mL of methanol was used to extract 100 mg of dried powdered material from transformed and untransformed *A. carvifolia* plants for 1 h at room temperature. The sonication bath was used for this purpose. A rota-evaporator was used at 40 °C to dry the supernatant after centrifuging the extract. Then, for the analysis of all cancer, these dried samples were dissolved in a 100% (v/v) DMSO solution at a final concentration of 40 mg/mL. For cytotoxicity assay, cells of cell lines were seeded in a 12-well plate and after 24 h of growth of cells, plant extracts were added to the medium in each well at a concentration of 200 ppm, and the viability was assessed after 48 h. All the conditions were run in triplicate. A control without drug treatment and a control where the cells were only treated with the solvent (DMSO) were included in all the assays. The viability of the cells was determined by the MTT [3-(4,5-dimethylthiazol-2-yl)-2,5-diphenyltetrazolium bromide.

### 2.11. Statistical Analysis

All the experiments were run in triplicate for analysis. The statistical significance level of the data were assessed by applying two-way ANOVA.

## 3. Results

### 3.1. DNA Barcoding for Identification of Plant

The chloroplast genomes *psbA-trnH* region, about 500 bp, was amplified successfully. To check the reliability of species-specific nucleotides, DNA samples were sequenced three times, with the same outcomes. For the purpose of confirming the plant species, the reference sequence with the GenBank Accession number (NCBI: FJ418751) was used. The sequence was identified as *psbA-trnH* of *A. carvifolia* after doing the CLUSTAL W in BioEdit program and BLAST in NCBI.

### 3.2. Transformation and Regeneration

*A. carvifolia* was successfully transformed with *A. tumefaciens* strain GV3101 carrying the *rol A* gene. A total of 400 explants were employed in each of the two separate transformation experiments. Although just five *rol A* transformants made it to maturity on the selection media, a 35% transformation efficiency was discovered. Morphological differences were depicted between transgenic plants carrying *rol A* gene and wild type plants in Figure 1. Dwarfness and harder texture of stems of transformed plants than wild type plants were also observed. *Rol A* transgenics had short, narrow, dark-colored leaves, and grew more quickly on the selection media.

### 3.3. Molecular Analysis

For the *A. carvifolia rol* gene transformants, PCR results revealed 308 bp for *rol A* gene and 781 bp for the *nptII* gene amplified products (Figure 2). The plasmid DNA of GV3101-PCV002-A yielded similar amplified products. Besides this, wild-type plants did not exhibit that these genes were present in their genome.

The incorporation of *rol* genes into the plant genome was confirmed by Southern blot analysis, which also provided information on the number of copies of each gene present in various transgenic lines. All five transgenic lines of *Artemisia carvifolia* showed the successful stable integration of *rol A* gene. One copy of *rol A* gene was seen in transgenic lines T1, T2, and T4. Whereas T3 and T5 showed two copies of the integrated gene as shown in Figure 3. Although there were differences in the expression level of these genes among studied lines, RT-PCR validated gene expression in all transformed lines. The *rol A* transgenic lines T3 and T5 in Figure 4 displayed the highest levels of expression due to double copy numbers as visible in Southern blot analysis results (Figure 3). β-actin was chosen as the positive control (housekeeping gene) which showed similar expression in all transgenic lines.

### 3.4. HPLC-DAD-Based Quantification of Flavonoids

Methanolic extracts of shoots of both transgenic and wild type *A. carvifolia* plants were prepared and then an HPLC-DAD system was used for the detection and quantification of flavonoids. The HPLC profile obtained was evaluated against the absorption spectra and retention duration of 10 standard compounds or flavonoid markers including vanillic acid, rutin, catechine, gallic acid, syringic acid, caffeic acid, coumaric acid, geutisic acid, ferulic acid, and cinnamic acid. Vanillic acid, syringic acid, gallic acid, coumaric acid, ferulic acid, caffeic acid, and cinnamic acid were present in both wild type plants and transformed plants but the concentration of these phenolic compounds was enhanced in *rol* gene transformants. As illustrated in Figure 5, unlike the wild type plants, flavonoids catechin and geutisic acid were discovered in the transformed plants only.

The vanillic acid content in wild type plants was 0.47 ug/mg DW but in the transformed plants it reached the highest 1.82 ug/mg DW in T3, showing a 4-fold increase. Rutin levels in wild type plants were 1.39 ug/mg DW, highest increase was observed in T5, showing a 2.7-fold increase of rutin. The gallic acid concentration was 0.62 ug/mg DW in wild type plants, increasing to 3.89 ug/mg DW with up to 6.2-fold in *rol A* transformants. Syringic acid wild type content was 1.05 ug/mg DW, increasing up to 3.7-fold to (3.90 ug/mg DW) highest in T5 transformed plants. The concentration of coumaric acid in wild type plants was 1.65 ug/mg DW but in transformed plants, it reached 2.85 ug/mg DW, showing the 1.7-fold increase in transformed plants. The concentrations of caffeic acid, ferulic acid, and cinnamic acid in wild type plants were 0.47 ug/mg DW, 1.39 ug/mg DW, and 1.67 ug/mg DW respectively and in transformed plants, these reached 2.60 ug/mg DW, and 5 ug/mg DW, and 4.23 ug/mg DW respectively showing a 5.5-fold increase of caffeic acid, a 3.6-fold increase of ferulic acid, a 2.56-fold increase of cinnamic acid in transformed plants. The results of HPLC showed that all flavonoids were increased up to the highest level in T3 and T5 transformed plants due to the integration of a double copy number of *rol A* gene.

Catechin and geutisic acid were absent in wild type plants and the concentration of catechin in transformed plants was 3.19 ug/mg DW and the amount of geutisic acid was 2.22 ug/mg DW in transformed plants. Statistical analysis was conducted, where the production levels of investigated phenolic compounds in *rol A* transgenic plants exhibited an extremely significant difference (*p* < 0.0001) when compared to the wild type *A. carvifolia* plants (Table 2).

### 3.5. Expression Analysis of Flavonoid Biosynthetic Pathway Genes through Real-Time qPCR

To evaluate the expression of two flavonoid biosynthetic genes (*PAL* and *CHS*), RT-qPCR was performed. In RT-qPCR, significant changes and higher expression degrees of the investigated genes of flavonoids biosynthetic pathway in plants transformed with *rol A* gene were observed. The expression of *PAL* and *CHS* genes was much low in wild type plants as shown in Figure 6. The *PAL* gene was highly expressed in the transformed plants as compared to *CHS* gene expression in transformed plants as an expression of *PAL* gene was 9–20-folds higher and expression of *CHS* gene was 2–6-fold higher in transformed plants. The *rol A* transgenic lines T3 and T5, which both carrying two copies of *rol A* gene, particularly showed higher expression of both *PAL* and *CHS* gene, with the highest expression in T3 line. These results showed that *rol A* gene plays role in inducing flavonoid biosynthesis by enhancing the expression of their biosynthetic genes, i.e., *PAL* and *CHS*.

### 3.6. Analyzation of the Antioxidant Potential of Rol Gene Transformed and Untransformed A. carvifolia

The antioxidant potential of *Artemisia carvifolia rol A* transgenic plants and wild type plants was assessed by using antioxidant assays. The total phenolic content and total flavonoid content was calculated as the equivalent of gallic acid (mg/g of DW) and quercetin (mg/g of DW) respectively, so TPC in wild type plants was 50 mg/g whereas *rol A* transgenic lines demonstrated an average increase of 1.4-fold in the phenolic content and the level of phenolic content was highest in line T3 (84 mg/g) followed by T5 (80 mg/g) as shown in Figure 7. TFC in wild type was 20 mg/g compared to 1–2-fold increase in *rol A* transgenic lines. Total antioxidant capacity was determined in terms of ascorbic acid equivalence (mg/g of the DW). TAC was increased up to an average 1–2-fold, as in wild type TAC was measured 40 mg/g but in transformed plants it was enhanced in all transgenic lines with highest level in T3 (60.7 mg/g) and then in T5 (50.9 mg/g). Similarly, total reducing power was also significantly increased in *rol A* transformed plants. In wild type plants, total reducing power was 70 mg/g. In transgenic line T1, T2, T3, T4, and T5 the TRP was measured as 99.4 mg/g, 100 mg/g, 130 mg/g, 97 mg/g, and 125 mg/g respectively, showing an average increase of 1.5–2-fold in total reducing power. *Rol A* gene showed extremely significant effect (*p* < 0.0001) on the antioxidant capacity of the *A. carvifolia* plants (Table 3 and Table 4). All these results indicated that the transgenic plant’s antioxidant capacity was increased by the incorporation of the *rol A* gene.

### 3.7. DPPH Free Radical Scavenging Assay

DPPH free radical scavenging is an efficient and widespread technique for determining the antioxidant activity of plant extracts. In this study, the DPPH assay was used to test the ability of transformed and untransformed plants to scavenge free radicals. According to results obtained, the *rol A* gene transformed plants showed significant increase in antioxidant potential as compared to untransformed wild type plants as shown in Figure 8. The extract of *rol A* transgenic line T3 displayed the maximum radical scavenging ability with an IC_50_ value of 206.9 ug/mL compared to the wild type plants having IC_50_ of 627 ug/mL. Similarly, the extracts of T1, T2, T4, and T5 exhibited more efficacy with less IC_50_ values (379, 450, 365.7, and 225.65 ug/mL respectively) as compared to wild type plants.

### 3.8. Measurement of Anticancerous Activity against Cancer Cell Lines through MTT Assay

The antiproliferative activity of untransformed and transgenic plants of *rol A* gene were checked by treating three different cell lines namely HeLA, MCF7, and HePG2 with the methanolic extracts of plants and thus the cell viability of all cell lines was evaluated. The results obtained showed that all transgenic lines were more effective against all cell lines as compared to wild type plant as shown in Figure 9. Mortality rate of Hela cells after treatment with untransformed *A. carvifolia* extract was 30% but after treating with *rol A* transgenic extracts, it increases up to 75%. Similarly, the mortality rate of MCF7 and HePG2 cells after treating with wild type plant extract was 32% and 36% respectively. After treatment with transgenic cell lines, the mortality rate of MCF7 and HePG2 was increased up to 70% and 74% respectively. Transgenic line T3 and T5 shows maximum effect on the viability of cancer cells. Thus, *rol A* gene showed extremely significant effect (*p* < 0.0001) on the enhancement of anticancerous properties of the plants under study (Table 5).

## 4. Discussion

Numerous investigations have demonstrated that *rol* genes are potent inducers or activators of the production of secondary metabolites in a variety of plant families [15]. After transformation with *rol A* gene, the morphological differences between transgenic plants carrying *rol A* gene and wild type plants were found. *Rol A* transgenics had short, narrow, dark-colored leaves and grew more quickly on the selection media. Similar leaf morphological changes were also reported in *Ajuga bracteosa* plant transformed with *rol A* gene in which leaves also showed moderate to severe wrinkling with epinasty [37]. Similarly, changes in leaf morphology due to *rol A* gene were also reported in tobacco plant [38,39,40,41]. Stunted growth of transformed plants than wild type plants was also observed which was similar to the findings already reported by researchers that *rol A* gene induce dwarfness in transformed plants of *Ajuga bracteosa*, potato, tomato, tobacco, and soybean plants [37,38,39,40,41]. All *rol A* soybean transformants showed significant variation in plant phenotype, primarily in terms of plant height, plant character, and leaf morphology. Additionally, a change in leaf morphology was noticed. The soybean plants that had undergone *rol A* transformation had elliptical-shaped leaves [42]. Similar findings were reported in which *Artemisia annua* plants that were transformed with *rol B* and *C* genes exhibited stunted growth and were shorter than those in the control group. These findings showed that while control plants had broad leaves, transformed plants had extremely short and narrow leaves. Compared to the control plants’ delicate stems, transgenic plant stems had a harder texture. In transformed plants, more branching was seen, but this had no effect on how quickly they grew overall [30]. Significant variations in growth and morphology were seen. *Agrobacterium tumefacians* mediated genetic transformation of *Artemisia carvifolia* plants with *rol B* and *rol C* genes have been reported but not with *rol A* gene. Morphological differences between wild type and *rol* genes transformants were reported by another group who used *rol B* and *rol C* genes for the genetic transformation of *Artemisia carvifolia* plants. They clearly mentioned the morphological heterogeneity between wild type or untransformed plants and plants transformed with *rol* genes. Broader leaves and more inflorescence were characteristics of *rol B* transformed plants, which developed more quickly on the selection media, as opposed to *rol C* transformed plants, which were resistant to regeneration and displayed a narrow leaf blade and shorter internodes [22]. In another study, Lettuce was genetically transformed with *Agrobacterium tumefaciens* strain GV3101 containing *rol C* gene. The construct was used to transform about 300 explants, with a transformation efficiency estimate of 65–75%; however, only three transgenic lines matured. All altered plants showed phenotypic changes in comparison to the control, including decreased leaf area, internodal lengths, stem heights, and inflorescence [43]. Additionally, *A. annua* and *A. dubia* plants that were genetically transformed with combined *rol ABC* genes also showed increased height and broad leaves [44], probably as a result of the combined action of *rol* genes. These morphological alterations may be the result of the hormonal imbalance brought on by the *rol* genes, which have been shown to induce abnormality in the ratio of auxin and cytokinin that favors cytokinins [45].

According to Southern blot experiment, transgenic lines T1, T2, and T4 all had one copy of the *rol A* gene. Contrarily, T3 and T4 displayed two copies of the integrated *rol A* gene. The results were similar to a study in which *rol A* gene transformants show single and double copy numbers [37]. HPLC analysis showed that all flavonoids are increased up to several folds and some flavonoids—such as catechin and geutisic acid—were absent in wild type plants but present in transformed plants. Similar effects of *rol* genes were observed previously [33] when the *rol B* and *rol C* transgenic plants were examined. A considerable rise in the content of identified flavonoids was seen. Rutin and caffeic acid levels increased up to 3-fold, quercetin levels increased by 4-fold and isoquercetin levels increased by 6-fold in the *rol B* transgenic plants. Contrarily, a threefold rise in rutin and quercetin, a 5-fold increase in isoquercetin, and a 2.6-fold increase in caffeic acid were seen in transgenics of the *rol C* gene [33]. *Rol* genes have been described as strong inducers of plant’s secondary metabolism [46]. Real time PCR confirmed that *rol A* gene play role in inducing flavonoid biosynthesis by enhancing the expression levels of *PAL* and *CHS*. The results correspond to the earlier reports in which the expression level of these both genes were higher in rol genes transformed *Brassica rapa* [47], *Artemisia carvifolia* [31], and *Lactuca serriola* [48] plants as compared to normal plants. Various reports describe that expression of PAL and CHS is directly related to the accumulation of flavonoids in the plant tissue. PAL enzyme catalyzes the flux of primary metabolites into the biosynthetic pathway of flavonoids through the phenylpropanoid pathway and hence performs a key role in flavonoid biosynthesis. CHS, the first enzyme of the flavonoid pathway, is an acyltransferase catalyzing the condensation of 4-coumaroyl CoA to the first flavonoid, naringenin chalcone, which is reported to be a rate-limiting step in flavonoid biosynthesis in different plants [33].

The antioxidant capacity of plants transformed with *rol A* gene were increased up to several folds. Similar findings were also shown in different reported researches on *Artemisia carvifolia* [31], *Artemisia annua* [30], *Lactuca serriola* L. [48], and *Lactuca sativa* L. [43] in which *rol* genes transformation boost the antioxidant potential in transformed plants. The transgenic lines of *rol A* gene displayed increase in the radical scavenging ability. The results are in accordance with the reported findings in which the *rol* genes transformation increased the antioxidant capacity of transformed *Artemisia carvifolia* plants.

Among secondary metabolites, flavonoids are famous for preventing or delaying cancer because it can prevent DNA mutations that takes place in important genes, such as oncogenes or tumor suppressor genes [49]. The results obtained were similar to the reported findings in which the viability of HeLA and MCF7 was 60% when treated with wild type *A. annua* extract and then decreased up to 40% and 35% respectively when treated with *rol B* transgenics [33]. Similarly, in previously reported research articles, some *Artemisia* species were found to be efficacious against the cancer cell lines like HeLA, P388 murine leukemia, and molt-4-human leukemia [50,51,52]. Moreover, cytotoxicity of *Artemisia annua* against MCF7, HepG2, and HelA cell lines was found significantly increased as compared to the wild type plant when transformed with *rol B* and *rol C* gene [33].

## 5. Conclusions

Secondary metabolites, especially flavonoids are the chemical building blocks of medicinal plants with clinically curative effects. To reap the amazing benefits of these metabolites on human wellbeing, genetic transformation plays very important role. The transformation of *Artemisia carvifolia* with *rol A* gene successfully improved the flavonoids content in the plant enhancing the antioxidant and anticancer potential of the plant.

## Figures and Tables

**Figure 1 metabolites-13-00351-f001:**
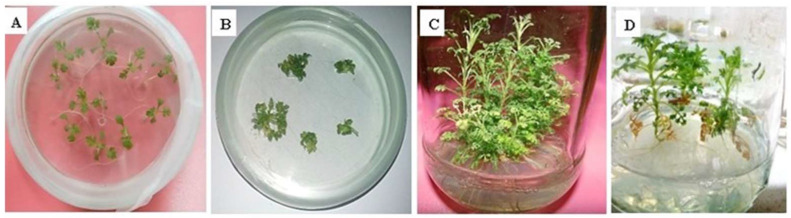
Genetic transformation of *Artemisia carvifolia*. (**A**) Seeds germination, (**B**) co-cultivation, (**C**) wild type plant, (**D**) *rol A* transgenic plant.

**Figure 2 metabolites-13-00351-f002:**
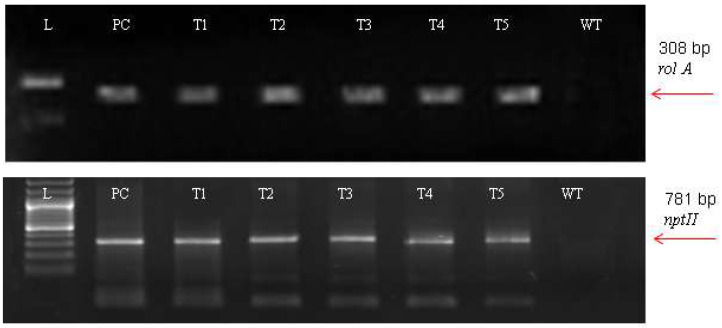
PCR result of transgenic *Artemisia carvifolia* harbouring *rol A* genes showing 308 bp for *rol A* and 781 bp for the *nptII* gene.

**Figure 3 metabolites-13-00351-f003:**
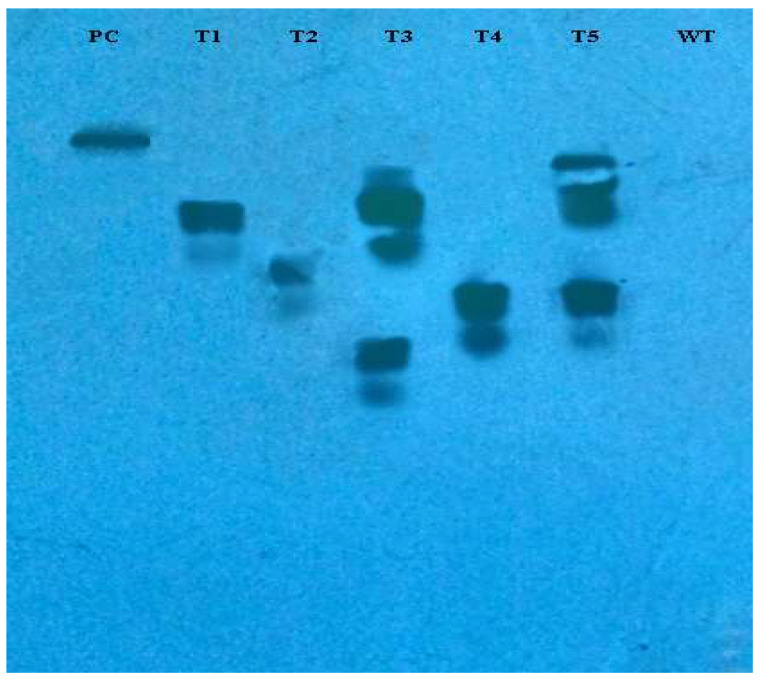
Southern blot analysis of PCR-positive plants demonstrating the incorporation of *rol A* gene into the *Artemisia carvifolia* genome. Transgenic lines T3 and T4 showed two copies of integrated gene.

**Figure 4 metabolites-13-00351-f004:**
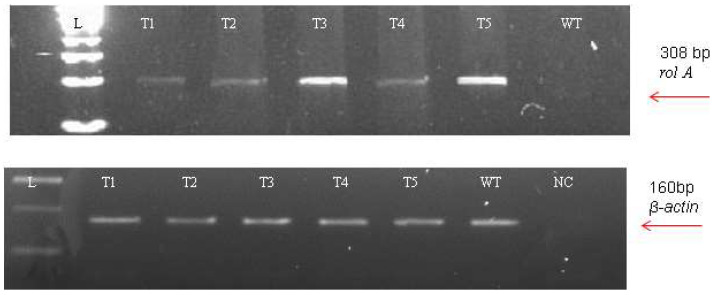
Reverse transcriptase PCR using the β-actin (160 bp) amplification as an internal control.

**Figure 5 metabolites-13-00351-f005:**
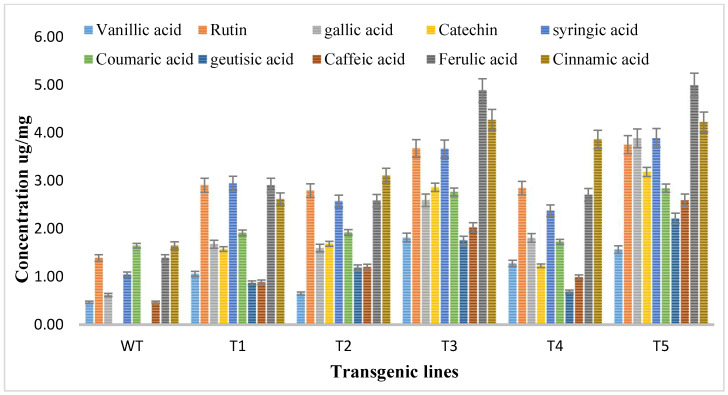
Quantitative analysis of flavonoids by HPLC in *A. carvifolia rol A* transgenic lines and wild type plants.

**Figure 6 metabolites-13-00351-f006:**
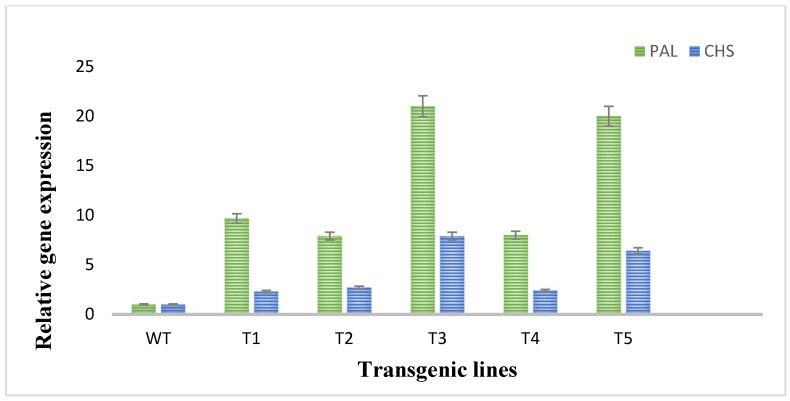
Quantitative real-time PCR analysis of flavonoid biosynthetic pathway genes. *PAL* and *CHS* stands for phenylalanine ammonia lyase and chalcone synthase respectively.

**Figure 7 metabolites-13-00351-f007:**
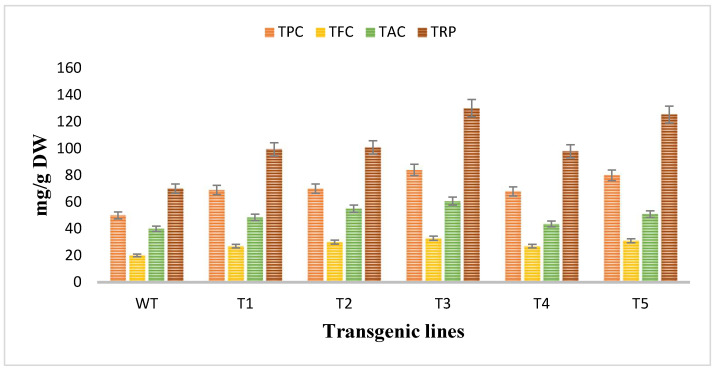
Evaluation of antioxidant potent through different antioxidant assays. TPC (total phenolic content, TFC (total flavonoid content), TAC (total antioxidant capacity), and TRP (total reducing power).

**Figure 8 metabolites-13-00351-f008:**
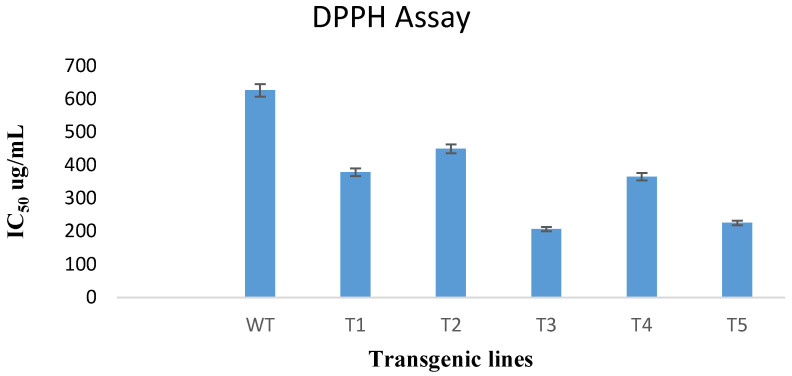
Results of the DPPH assay of extract of five *rol A* integrated transgenic lines and wild type plant extract (WT).

**Figure 9 metabolites-13-00351-f009:**
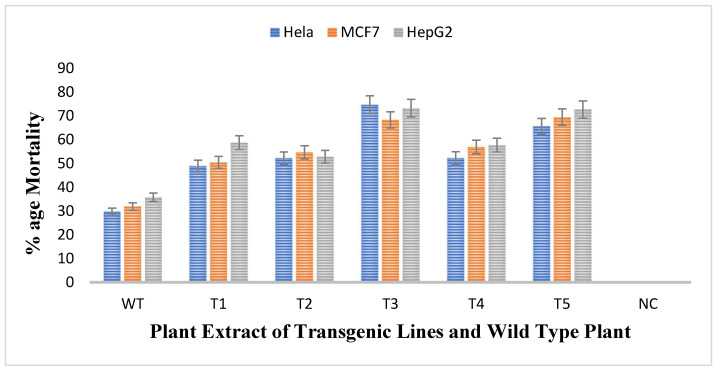
MTT assay for determination of cytotoxic activity of plant extracts (40 mg/mL) against HeLA, MCF7, and HePG2 cancer cell lines.

**Table 1 metabolites-13-00351-t001:** Retention time of examined flavonoids with wavelength.

S. No.	Standard Flavonoids	Signal Wavelength (nm)	Retention Time (min)
1	Vanillic acid	257	9.116
2	Rutin	257	12.649
3	Gallic acid	279	3.679
4	Catechin	279	7.003
5	syringic acid	279	9.76
6	Coumaric acid	279	13.79
7	Geutisic acid	325	7.433
8	Caffeic acid	325	9.252
9	Ferulic acid	325	12.642
10	Cinnanic acid	325	13.79

**Table 2 metabolites-13-00351-t002:** Two-way ANOVA for HPLC analysis.

Source of Variation	Degrees of Freedom	Sum of Squares	Mean Square	F-Value	Prob.
Polyphenols	9	117.4	13.04	54.64	0.0000
Transgenic lines	5	120.2	24.04	1592	0.0000
Interaction	45	20.15	0.4478	2933	0.0000
Residual (error)	120	0.9835	0.008196		
Total	179	258.7			

**Table 3 metabolites-13-00351-t003:** Two-way ANOVA for antioxidant assays.

Source of Variation	Degrees of Freedom	Sum of Squares	Mean Square	F-Value	Prob.
Interaction	15	3330	222.0	68.68	0.0000
Transgenic lines	5	7898	1580	488.8	0.0000
Antioxidant assays	3	56270	18760	5804	0.0000
Residual (error)	48	155.1	3.232		
Total	71	67660			

**Table 4 metabolites-13-00351-t004:** Two-way ANOVA for DPPH free radical scavenging assay.

Source of Variation	Degrees of Freedom	Sum of Squares	Mean Square	F-Value	Prob.
Interaction	15	472.6	31.51	17.32	0.0000
Transgenic lines	5	3281	656.2	360.7	0.0000
Concentrations	3	22130	7376	4055	0.0000
Residual (error)	48	87.33	1.819		
Total	71	25970			

**Table 5 metabolites-13-00351-t005:** Two-way ANOVA for MTT assay.

Source of Variation	Degrees of Freedom	Sum of Squares	Mean Square	F-Value	Prob.
Cancer cell lines	2	202.2	101.1	118.2	0.0000
Transgenic lines	6	34110	5684	6645	0.0000
Interaction	12	291.4	24.28	28.38	0.0000
Residual (error)	42	35.93	0.8554		
Total	62	34640			

## Data Availability

Data is not publicly available due to privacy.

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
