# Peer review of "Enhanced Antioxidant and Anticancer Potential of Artemisia carvifolia Buch Transformed with rol A Gene"

_metabolites, 2023, doi:10.3390/metabo13030351_

Round 1

Reviewer 1 Report

Appreciable presentation.

Author Response

No comments were found.

Reviewer 2 Report

Authors proved, that transformed plants display significantly higher polyphenol context, but transformed plant are shorter and probably with slower growth rate. It is possible to compare the amount of obtained polyphenols with regard to the plant and the nutrients used (also with regard to costs)?

It is was possible determine chelation ability of polyphenol extracts, or their effect on the Fenton reaction?

Figure 9 -applied extract dose is not defined.

In the experimental section, determination of anticancer activity should be described more detailed.

Author Response

  1. Authors proved, that transformed plants display significantly higher polyphenol context, but transformed plant are shorter and probably with slower growth rate. It is possible to compare the amount of obtained polyphenols with regard to the plant and the nutrients used (also with regard to costs)?

Answer: As mentioned in lines 261-263, although the transformed plants have dwarfness but they grew quickly on selection media as compared to wild type plants. So, the increased amount of polyphenols within short time period can be obtained through transformation.

  1. It is was possible determine chelation ability of polyphenol extracts, or their effect on the Fenton reaction?

Answer: All the adopted methodology was standard reported protocols. No such possibility is reported yet.

  1. Figure 9 -applied extract dose is not defined.

Answer: Applied extract dose has been mentioned in figure 9.

  1. In the experimental section, determination of anticancer activity should be described more detailed.

Answer: The methodology of anticancer activity has been described in detail (line 232-243).

Reviewer 3 Report

Dear Editors,

Thank you for the opprotunity to revise article „Enhanced Antioxidant and Anticancer Potential of Artemisia Carvifolia Buch Transformed with Rol A Gene”.

The article is very interesting. In the study, flavonoids were detected in the Artemisia carvifolia Buch wild type and then flavonoid content was enhanced by genetically modifying A. carvifolia with rol A gene. Aditionally, antioxidant potential and cytotoxicity were also tested.

The article is well written. The introduction section is quiet short and might be improved.

The material and methods section is very detailed and there is no need to change anything.

The results are clearly demonstarted and the discussion is rigorous.

The conclusions section must be included and clearly demonstrated.

Author Response

  1. The article is well written. The introduction section is quite short and might be improved.

Answer: The introduction section has been enhanced, highlighted yellow.

  1. The conclusions section must be included and clearly demonstrated.

Answer: Conclusion has been included.

Reviewer 4 Report

The novelty and the quality of the manuscript are good and it does not need extensive improvement before publication. It is carefully organized and written. It is easy to follow it and contains clear comments and conclusions.  In my opinion, this manuscript is very detailed and meticulous, it covers all the literature in the field with critical point of view. The topic have been completely covered and is well connected through the text. There is a significant  novelty in presented topic.  For all these reasons, I can recommend the acception of the manuscript after minor revision:

 1. I think that part 3.2. Transformation and Regeneration could be extended, more examples  should be added. This would be valuable for later publication citation.

 2. The superiority of  the g the genetic transformation of Artemisia carvifolia Buch than other transformation of this plant should be more emphasized.

 3. The manuscript should be extended in scientific discussion. The authors presented their results and compared to some works, but did not present explanations for the reasons to reach these results.

 4. Not all of the described results are covered in the discussion section.

 5. No all information was given of activity of Artemisia species against the cancer cell lines like HeLA

Author Response

  1. I think that part 3.2. Transformation and Regeneration could be extended, more examples should be added. This would be valuable for later publication citation.

Answer: More examples have been added in line 261-263.

  1. The superiority of the genetic transformation of Artemisia carvifolia Buch than other transformation of this plant should be more emphasized.

Answer: Plant extracts attained from Artemisia species are used for the treatment of depression, anxiety, insomnia, epilepsy, stress, irritability and psychoneurosis. Additionally, plants from Artemisia species hold a wide range of biological functions like antibacterial, antitumor, hepatoprotective, antimalarial, antiseptic, antispasmodic, and antirheumatic activities (lines 45-55). The level of flavonoids in Artemisia plants is relatively low, and fluctuations have also been observed in the same plant at different growth stages (lines 67,70). In order to improve the synthesis of secondary metabolites in plants, genetic engineering has proven to be the most effective method among various approaches. It has been revealed in different studies that rol genes act as a powerful inducers or activators of secondary metabolite production in various plant families (lines 73-77). There has been extensive research on rol B and rol C genes transformation in Artemisia carvifolia and their role in enhancement of secondary metabolites has been observed but rol A gene has not been examined in this regard, previously.

  1. The manuscript should be extended in scientific discussion. The authors presented their results and compared to some works, but did not present explanations for the reasons to reach these results.

Answer: Discussion added, lines 443-445, 462-469, 484-486.

  1. Not all of the described results are covered in the discussion section.

Answer: In discussion, transformation and regeneration (lines 406-445), southern blotting (lines 446-449), HPLC (lines 449-458), RT PCR (lines 458-462), measurement of antioxidant effects (lines 463-466), DPPH assay (lines 466-461), MTT assay (lines 472-477).

  1. No all information was given of activity of Artemisia species against the cancer cell lines like HeLA

Answer: The activity of Artemisia species against cancer cell lines like HeLA have been added (line 484-486).